# Interactive Cognitive Motor Training: A Promising Approach for Sustainable Improvement of Balance in Older Adults

**Longhai Zhang, Jiawei Guo, Jing Zhang, Ling Zhang, Yanbing Li, Shutong Yang, Wenfei Zhu \*** and **Fei Guo \***

School of Physical Education, Shaanxi Normal University, Xi'an 710119, China; hae@snnu.edu.cn (L.Z.); guojiawei729@snnu.edu.cn (J.G.); zhangjiing0578@snnu.edu.cn (J.Z.); zhangling1@snnu.edu.cn (L.Z.); liyanbing@snnu.edu.cn (Y.L.); yangshutong123@snnu.edu.cn (S.Y.)

\* Correspondence: wzhu@snnu.edu.cn (W.Z.); guofei@snnu.edu.cn (F.G.); Tel.: +86-177-9232-1530 (W.Z.); +86-133-8498-5341 (F.G.)

**Abstract:** Physical exercise has been shown to improve balance, gait, and cognitive function in older adults. Interactive cognitive-motor training (ICMT) combines physical exercise and cognitive stimulation, but few studies have focused on the effect of ICMT on static and dynamic balance in older adults. This study aims to improve the balance of older adults to reduce fall-related injuries for sustainable development goals. We randomly assigned 38 older adults to either the ICMT group ($n$ = 22) or the control group ($n$ = 16). The ICMT group participated in 60-min exercise sessions three times a week for 12 weeks, while the control group maintained their regular activities. The static and dynamic balances were assessed before and after the intervention. The results indicated the ICMT group demonstrated significant improvements in static balance, specifically in swing path and velocity (V) in the medial-lateral (M-L) direction with eyes open ($p < 0.05$). With eyes closed, the ICMT group showed significant improvements in total swing path and area compared to the control group ($p < 0.05$). In dynamic balance, there was a significant difference in the lateral (L) angle limit of stability (LOS) between the ICMT group and the control group ($p < 0.05$). In conclusion, ICMT effectively enhances static balance and maintains dynamic balance in older adults.

**Keywords:** sustainable health; exergame; static balance; dynamic balance

## 1. Introduction

The United Nations emphasizes that the assurance of healthy lifestyles and the promotion of well-being across all age groups are imperative for sustainable development [1]. The WHO Healthy Cities Project affirmed the inseparable connection between human health and sustainable development [2]. The European Union also set out its approach to identifying the best plans for promoting the health of older adults and building a sustainable development model for healthy aging [3]. According to the United Nations Population Division, the proportion of the global population aged 65 years or older was 10% in 2022 [4]. Data from China's seventh census shows 18.70% of China's population is aged 60 and over, and 13.50% is aged 65 and over [5]. Research shows that falls have become the primary cause of injury deaths among people aged 65 and over [6]. Falls in older adults can easily lead to other illnesses such as death, fractures, and traumatic brain injuries. It also generates psychological stress and a loss of independence [7]. Care and rehabilitation add significantly to the financial strain [6,8].

Factors that contribute to falls in older adults may include balance, cognitive function, and gait [9–11]. Dance [12], Tai Chi [13], and strength training [14] are effective in improving balance in older adults. Some studies indicate interactive cognitive-motor training (ICMT) can improve balance [10]. ICMT is an intervention that integrates cognitive and motor tasks to promote an individual's cognitive function and motor skills [15]. ICMT is more enjoyable than traditional physical exercise, offers more cognitive engagement and real-time feedback [16], and has been shown to improve gait and balance in older adults

while improving cognitive function [7]. ICMT, combining virtual reality, motion sensing technologies, and interactivity, represents an innovative technological solution in the field of sustainable development [17]. ICMT can be constantly updated and iterated as technology advances, from the initial exercise following videos to today's somatosensory control, constantly reinforcing the combination of exercise and cognition, thus achieving the concept of sustainable development of entertainment and health [18]. The dual-task intervention theory suggests that combining cognitive tasks with balance exercises enhances coordination and response abilities [19]. However, few studies have examined the effect of ICMT on static and dynamic balance in older adults.

In this study, we conducted a 12-week randomized controlled trial to investigate the long-term effect of ICMT on balance in older adults. The research primarily aims to examine the potential of ICMT in enhancing balance capabilities among older adults, thereby making a significant contribution to sustainable social well-being and the efficient management of healthcare resources. We hypothesized that ICMT can significantly improve the balance of older adults.

## 2. Materials and Methods

### 2.1. Participants

Our study was conducted in June 2022, and recruitment was carried out through distributing flyers in the community and organizing online informational sessions. The experiment took place in August 2022, with participants wearing accelerometers for one week before the experiment. Pre-test data collection included gathering participants' demographic information and conducting tests for static and dynamic balance. The exercise intervention phase took place in September 2022 and lasted for 12 weeks. The post-test was conducted in November 2022. In this study, 60 volunteers were recruited from the surrounding community of Xi'an, Shaanxi Province, China, with the following participant inclusion criteria: (1) Measures of cognitive function were used in the Mini-mental State Examination (MMSE) questionnaire to ensure that the participant has normal cognitive function [20,21]. (2) The Physical Activity Readiness Questionnaire (PAR-Q) was used to investigate whether the participants had a movement-limiting-related disorder [22]. The Physical Activity Readiness Questionnaire (PAR-Q) is a widely used, self-administered questionnaire designed to assess an individual's readiness to engage in physical activity. It aims to identify individuals with underlying health issues that could be aggravated by exercise. The PAR-Q is commonly used by fitness professionals, health practitioners, and exercise facilities as a screening tool to ensure the safety of individuals before they begin a new exercise program. The native language version of the questionnaire has been proven to be valid [23]. (3) Normal visual and auditory function and no motor impairment in the last six months. The exclusion criteria are as follows: (1) cognitive dysfunction; diabetes mellitus and neuropathy or peripheral artery disease; (2) movement disorders caused by orthopedic or neurological diseases; other sports disorders that hinder participation in training; (3) drop-out due to personal or objective reasons, resulting in the inability to complete the intervention or post-test.

Participants were assigned to either the ICMT group or the control group using computer-based random sorting and grouping, with each participant being numbered and then randomly assigned to one of the two groups using SPSS (Statistical Package for Social Sciences, Inc., Chicago, IL, USA). After applying the inclusion criteria, 56 participants met the eligibility requirements and were enrolled in the study. All participants were randomized into an ICMT group ($n = 28$) and a control group ($n = 28$). After 12 weeks, for the ICMT group, four participants withdrew from the intervention for medical reasons (infected with COVID-19: $n = 3$, developed arthritis: $n = 1$); two participants withdrew from the intervention for time conflicts (did not complete the post-test: $n = 1$, did not attend more than 20% of training: $n = 1$). For the CG, six participants withdrew from the intervention for medical reasons (infected with COVID-19: $n = 4$, lower limb injury: $n = 2$); six participants withdrew from the intervention for time conflicts (did not complete the post-test: $n = 6$).

Leaving a total of 38 participants (ICMT group = 22, control group = 16). This study received ethical approval from Shaanxi Normal University (202116003), and signed informed consent was obtained from all participants. We collected information on participants' age, sex, education level, smoking status, and alcohol consumption. We also measured each participant's light physical activity (LPA) and moderate-to-vigorous physical activity (MVPA) using an accelerometer-based activity monitor (ActiGraph WGT3X-BT, Pensacola, FL, USA). Participants were asked to wear the accelerometer on their non-dominant wrist for seven consecutive days. These seven days were worn before the intervention to prevent participants' prior physical activity from influencing the results, and we analyzed physical activity as a control variable [24]. This study selected the proportion of total wear time spent in MVPA (MVPA%) as a covariate. It is important to note that the study has both strengths and limitations. The strengths include a rigorous randomized controlled trial design, objective measures of physical activity and balance, and comprehensive inclusion and exclusion criteria. The integration of real-time heart rate monitoring and the systematic progression of game difficulty enhanced the study's methodological robustness. However, there are limitations to consider. The relatively small sample size could potentially limit the generalizability of the findings. Additionally, attrition during the intervention period and the potential influence of participant dropout on outcomes should be acknowledged.

Figure 1 shows a flow diagram of participants' recruitment and retention. The baseline characteristics showed no significant differences between the two groups (Table 1).

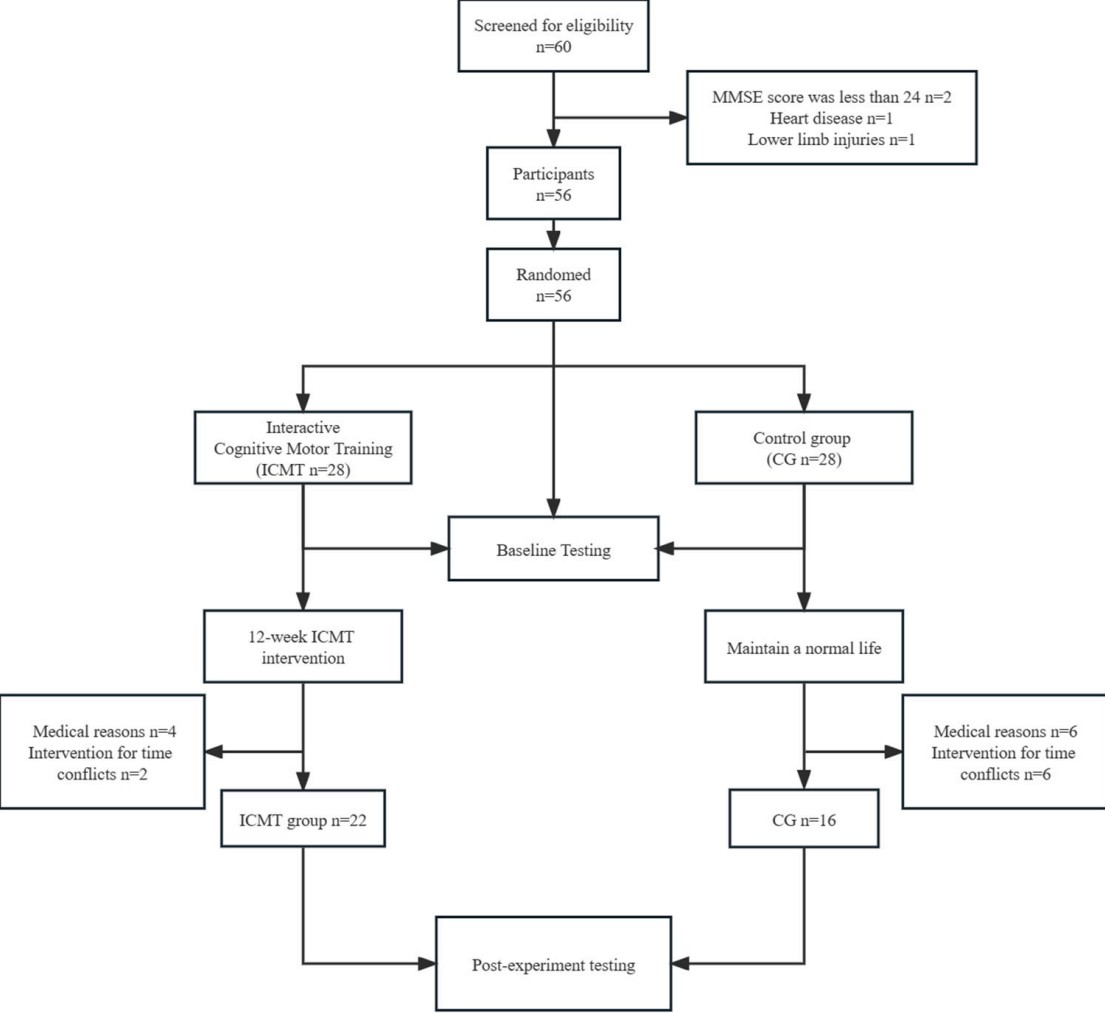

**Figure 1.** Flowchart for participants.

**Table 1.** General characteristics of the study participants.

| | | ICMT (*n* = 22) | | CG (*n* = 16) | | Total | | F/$\chi^2$ | *p* |
|---|---|---|---|---|---|---|---|---|---|
| Age M, (SD) | | 65.6 | 4.2 | 65.8 | 3.2 | 65.6 | 3.8 | 0.005 | 0.945 |
| Sex (%) | Men | 8 | 14% | 6 | 11% | 14 | 37% | 0.008 | 0.929 |
| | Women | 14 | 25% | 10 | 18% | 24 | 63% | | |
| Education (%) | Primary school | 1 | 2% | 1 | 2% | 2 | 5% | | |
| | Junior high school | 8 | 14% | 2 | 4% | 10 | 26% | | |
| | Senior high school | 5 | 9% | 7 | 13% | 12 | 32% | 0.982 | 0.328 |
| | Junior college | 6 | 11% | 2 | 4% | 8 | 21% | | |
| | Undergraduate | 2 | 4% | 4 | 7% | 6 | 16% | | |
| Smoking (%) | Yes | 1 | 2% | 1 | 2% | 2 | 5% | 0.051 | 0.822 |
| | No | 21 | 38% | 15 | 27% | 36 | 95% | | |
| Alcohol drinking (%) | Yes | 4 | 7% | 1 | 2% | 5 | 13% | 1.128 | 0.295 |
| | No | 18 | 32% | 15 | 27% | 33 | 87% | | |
| MMSE, M (SD) | | 29.0 | 1.1 | 28.7 | 1.1 | 28.8 | 1.1 | 0.786 | 0.381 |
| MVPA%, M (SD) | | 16.9 | 5.4 | 18.3 | 7.0 | 17.5 | 6.0 | 0.535 | 0.469 |

Note: ICMT—Interactive cognitive-motor training; CG—Control group; M ± SD—Mean ± Standard Deviation; MVPA—moderate-to-vigorous physical activity.

### 2.2. Intervention

Our intervention training spanned 12 weeks, with three weekly sessions each lasting 60 min, incorporating a 10-min warm-up and a 5-min cool-down period. During the exercise period, participants wore Fitmao heart rate armbands, and their heart rates were monitored in real-time using the Fitmao Heart Rate System (Fitmao, Shenzhen Fitmao Fitness Technology Co., Ltd., Shenzhen, China) [25]. Participants were ensured to exercise at a moderate to vigorous intensity (exercise intensity = 65–75% HRmax). Before the formal intervention, the researchers had provided participants with a comprehensive explanation of the game rules and conducted specialized basic movement training for each game. This ensured that older adults could effectively adapt to the exercise intensity of somatosensory games and successfully achieve their training goals through the skillful execution of exercise interventions. During the formal intervention, participants were divided into four teams, with the game being changed on a weekly basis. Each stage lasted for four weeks, resulting in a total of three stages. The difficulty of the game increased progressively in each stage, considering the participants' training level, to prevent diminishing intervention effects as their skills improved. Participants were encouraged to increase the exercise intensity if their heart rate remained low. Conversely, if the heart rate exceeded the moderate intensity range, participants were instructed to rest or reduce their exercise intensity to maintain a moderate-intensity level of heart rate. In summary, the cornerstone of our exercise session design is individualization. Irrespective of gender, we tailor exercise intensity, training volume, repetitions, sets, and rest intervals to meet the needs and objectives of each participant, ensuring optimal health and fitness outcomes.

Interactive Cognitive-Motor Training

The game devices used by the ICMT group in this study were Switch (Nintendo Co., Ltd., Kyoto, Japan) and Joy-Con™ (See Figure 2). The Nintendo Switch gaming system has been proven effective in various fields, such as improving muscle strength in older adults [26] and enhancing motor learning abilities in stroke patients [27]. However, its application to improving balance among older adults has not been fully explored. Our study represented the first instance of utilizing this ICMT program specifically for addressing balance improvement in older adults. Forty-five-minute group sessions were arranged three times a week for 12 weeks. The following three games were selected for this study:

(1) Fitness boxing 2

This is a boxing virtual game. During training, the participant swung the Joy-Con™ in response to on-screen prompts and learned basic boxing moves and sets, including straight punches, swinging punches, hooks, dodges, dive dodges, U-bends, and steps. Participants

were required to make judgments based on the movement markers, to deliver precise strikes when the markers moved into the swing box, and to adjust their movements based on the instant evaluation (Figure 3A,B).

(2)    ZUMBA

ZUMBA Fitness Dance is a cardio fitness game that incorporates a variety of dance styles. Participants imitated the dance moves of the instructor on the screen with a Joy-Con™, and the device scored and evaluated the moves tracked by the Joy-Con™ in real-time, allowing participants to correct their moves based on the evaluation. At the end of the dance, an overall score was given and a calorie count was displayed to motivate participants to exercise (Figure 3C,D).

(3)    Mario Tennis Ace

Mario Tennis Ace is a tennis tournament game. This study used a 'swing mode' intervention that required participants to use randomly occurring 'opportunity areas' or 'smash' on the court to win. The participant swayed the Joy-Con™ to hit the ball while moving to control the position and direction of the character and adjust their movements according to the system's instantaneous evaluation of each ball. The game required two participants to intervene at the same time (Figure 3E,F).

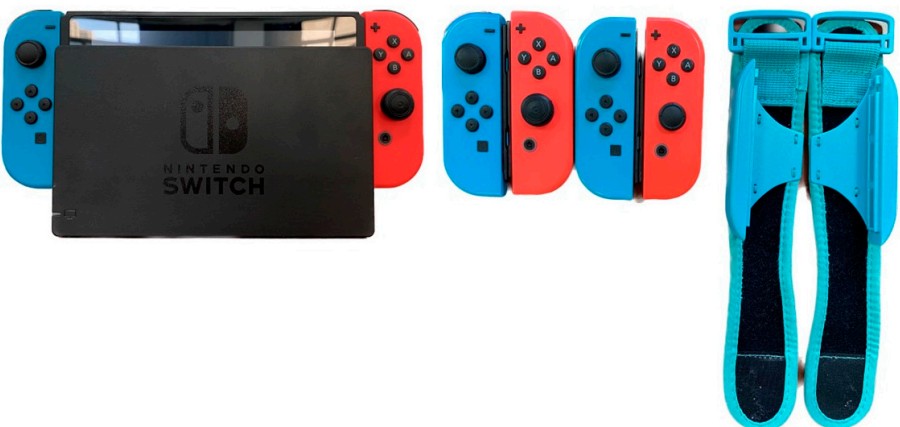

**Figure 2.** Nintendo Switch ICMT system.

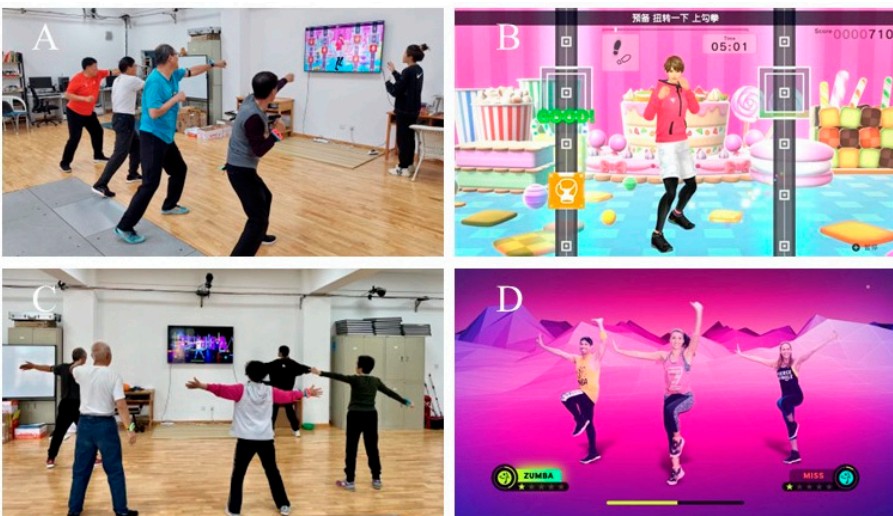

**Figure 3.** *Cont.*

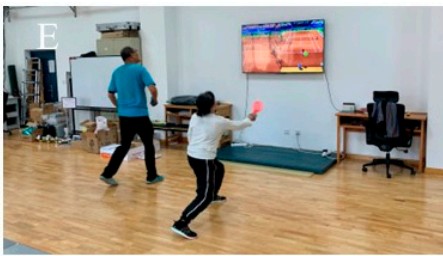
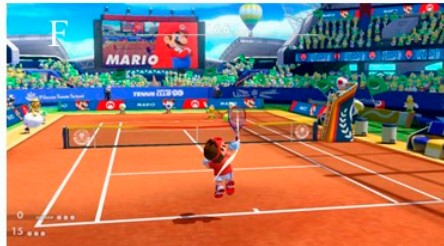

**Figure 3.** Process of ICMT intervention for older adults. (**A**) Older adults training with Aerobic Boxing 2; (**B**) diagram of the Aerobic Boxing 2 interface; (**C**) older adults training with Zumba; (**D**) diagram of the Zumba interface; (**E**) older adults training with Mario Tennis; (**F**) diagram of the Mario Tennis interface.

## 3. Balance Test

### 3.1. Static Balance Test

A three-dimensional force platform (Kistler, 9287CA, Winterthur, Switzerland) and MARS balance analysis software were used. "MARS" was measurement, analysis, and reporting software. Combining with the force platform (Kistler, 9287CA, Winterthur, Switzerland), the static balance of the participants could be measured and analyzed precisely, and the reliability has been proven [28,29]. The *X*-axis of the table is defined as parallel to the frontal plane in the medial and lateral directions, the *Y*-axis is perpendicular to the frontal plane in the A and P directions, and the origin is the center of the table. The MARS software offers three footwear options: barefoot, running shoes, and high heels. For this study, the "running shoes" option was selected. The metrics tested in this study included the total sway path length (mm), which is the length of the center of pressure (COP) sway trajectory calculated by summing the point-to-point Euclidean distances. The sway path A-P refers to the COP sway trajectory in the anterior-posterior (A-P) direction, while the sway path M-L refers to the COP sway trajectory in the medial-lateral (M-L) direction. The total length of the sway velocity (V) (mm/s) is the common length of the COP sway trajectory calculated by summing the point-to-point Euclidean distances and dividing by the measurement time. The sway velocity in the A-P direction is referred to as sway V A-P, while the sway velocity in the M-L direction is referred to as sway V M-L. The total sway area (mm$^2$) is the area swayed by the COP trajectory relative to the central standing point, calculated as the ellipse containing 95% of the COP. The sway area A-P (mm $\times$ s) is the time integral of the A-P component of the COP concerning the mean value of the A-P component. The sway area M-L (mm $\times$ s) is the time integral of the M-L component of the COP concerning the mean value of the M-L component. It should be noted that this direction-specific area is obtained by taking the time integral and therefore has units of mm $\times$ s, not mm$^2$.

Previous studies have demonstrated that a commonly used duration for balance testing ranges from 10 to 30 s. In this study, a testing duration of 20 s was selected. During the preliminary experiments conducted before the actual study, it was observed that individuals exhibited varying degrees of fatigue recovery, generally ranging from 30 to 60 s. Therefore, a resting duration of 60 s was chosen [14,28,29]. During the test, participants stood with their feet shoulder-width apart and their waist and abdomen straight. The participant performed the test three times with eyes open and closed for 20 s each, with a recovery interval of 60 s to prevent fatigue between tests. The lower the value of each indicator collected, the more stable the participant is. This test was performed three times, and the average value of static balance ability was measured. In addition, we asked participants to wear the same shoes for the pre-test and post-test (Figure 4).

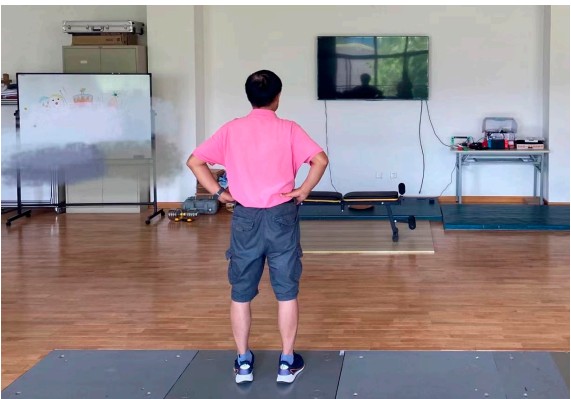

**Figure 4.** Schematic diagram of the static balance test.

*3.2. Dynamic Balance Test*

The Biodex Balance System was used to measure dynamic balance (Biodex System SD with V4.x software, Biodex Medical Systems, Inc., New York, NY, USA), which has been used before [30]. The tests include the limits of stability (LOS) test and the postural stability test. Previous studies have revealed that stability limits and postural control are good indicators for assessing dynamic balance. In pre-experiments, we found that the difficulty, safety, and intensity of these two tests were more suitable for older adults. When the platform is static, the system determines the position of the center of gravity (COG) from the participant's height (55% of height) and thus measures the angular shift of the participant's COG during the test, e.g., the LOS test. When the test platform is dynamic, the system measures the participant's ability to control balance by measuring the difference between the angle of the platform and the horizontal position of the platform, or the degree of shift over time, e.g., postural stability test.

Before the test began, the participants' age, sex, height, and weight were entered into the test system to create a profile, and the test was explained to the participant before the formal testing session. The participants were asked to stand on the test platform with their arms naturally down on either side of their torso and their left and right heels at (F,8) and (F,14), respectively, with their toes 30° apart (the line between the heel and third toe of the left and right feet was parallel to the 15° line on the test platform). The test was set for 20 s and repeated three times, with a 20 s break at the end of each test to avoid fatigue. Participants were not allowed to bend or flex their knees during the test and were required to retest if they stepped on or held on to the handrail (Figure 5).

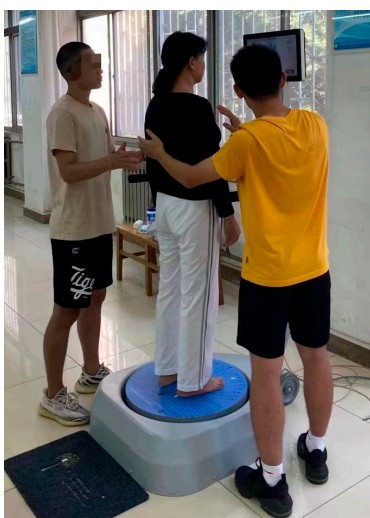

**Figure 5.** Schematic diagram of the dynamic balance test.

**(1). Limits of stability (LOS) test:** Tests the participant's ability to move and control their COG in a plane. After completing a target path, the participant returned the cursor to the center and held it there for 3 s until the next target path appeared. Higher values of the results indicated the participant was better able to stabilize the limits. The test metrics were average angle, anterior angle, P angle, M angle, L angle, M-A angle, L-A angle, M-P angle, and L-P angle (See Figure 6).

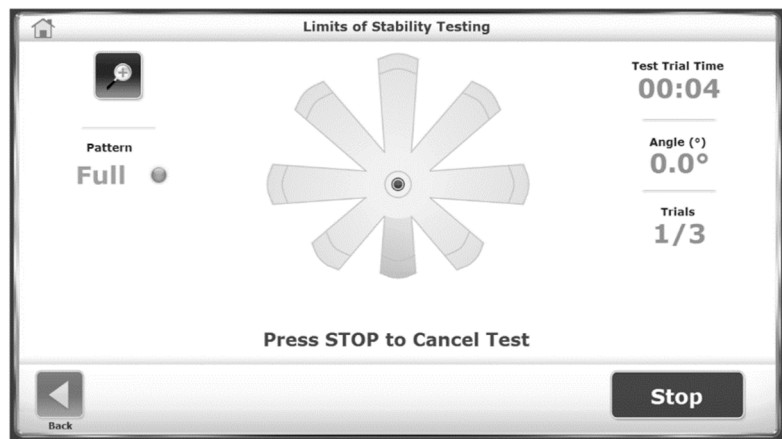

**Figure 6.** Limits of stability test.

**(2). Postural stability test:** During each trial, participants maintained their position to keep the cursor in the center. A 10 s countdown was displayed on the screen after each trial to allow for rest before the next trial. The stability index and sway index were used to indicate the participants' postural stability. A smaller value of the stability index indicates less deviation of the participant's COG, while the sway index refers to the degree of swaying of the participant on the test platform, with a smaller value indicating less body swaying. Test indicators included the total stability index, anterior-posterior directional stability index, left-right directional stability index, overall sway index, anterior-posterior directional sway index, and left-right directional sway index. (Figure 7).

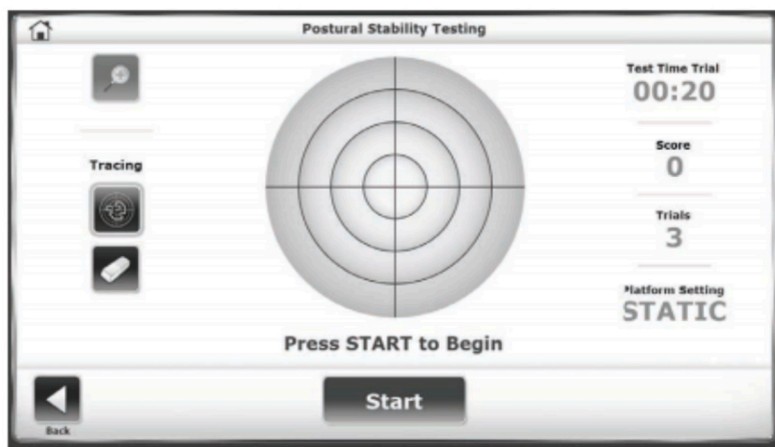

**Figure 7.** Postural stability test.

*3.3. Statistical Analysis*

SPSS 25.0 (Statistical Package for Social Sciences, Inc., Chicago, IL, USA) was employed for all data analyses. Descriptive statistics were presented as Mean(M) ± Standard Deviation (SD) for continuous parameters and percentages for categorical parameters. A one-way ANOVA or chi-square test was used to test the differences in the baseline values between the two groups. An analysis of covariance was used to examine the differences between groups before and after the intervention, controlling for sex, age, education, and

MVPA%. Significance levels were set at $p < 0.05$. The effect size was calculated via $\eta_p^2$ to assess the importance of the measured changes. The magnitude of $\eta_p^2$ values was classified as no effect size ($0 \leq \eta_p^2 < 0.01$), small effect size ($0.01 \leq \eta_p^2 < 0.06$), medium effect size ($0.06 \leq \eta_p^2 < 0.14$), and large effect size ($\eta_p^2 \geq 0.14$) [31] Paired samples t-tests were used to analyze variables that have significant differences between groups in the analysis of covariance. A power analysis conducted with G*POWER 3.1 (Universitat Kiel, Dusseldorf, Germany) determined that 52 participants were needed in this study for a power of 0.80, an effect size of 0.4 and $\alpha = 0.05$, a grouping of 2, and a number of covariates of 5 [32,33]. An additional 8 older adults were recruited for this study to account for sample attrition.

## 4. Results

Table 2 shows the differences in static balance in the eyes between the open and closed states of the older adults in the two groups. After 12 weeks of intervention, for closed-eye static balance, compared to the CG, we found significant improvement in sway path M-L ($F = 4.299$, $p = 0.047$, $\eta_p^2 = 0.122$), and sway V M-L ($F = 4.269$, $p = 0.047$, $\eta_p^2 = 0.121$) for the ICMT groups, with a medium effect size. There was a medium effect size for the total area of swing ($F = 3.100$, $p = 0.088$, $\eta_p^2 = 0.091$).

**Table 2.** Differences in static balance in the eyes open and closed states of the older adults in the two groups.

| | ICMT ($n$ = 22), M $\pm$ SD | | | | CG ($n$ = 16), M $\pm$ SD | | | | Comparison of Two Groups | | | |
| --- | --- | --- | --- | --- | --- | --- | --- | --- | --- | --- | --- | --- |
| | Pre | | Post | | Pre | | Post | | F | $p$ | $\eta_p^2$ | 1-$\beta$ |
| | | | | (Open Eyes) | | | | | | | | |
| The total length of the swing [mm] | 209.8 | 51.8 | 205.1 | 60.4 | 213.4 | 47.7 | 226.1 | 49.6 | 0.922 | 0.344 | 0.029 | |
| Sway path A-P [mm] | 176.2 | 48.4 | 177.0 | 59.2 | 181.7 | 48.4 | 195.5 | 49.1 | 0.660 | 0.423 | 0.021 | |
| Sway path M-L [mm] | 81.8 | 17.8 | 72.4 | 13.7 | 79.7 | 9.8 | 80.4 | 12.1 | 4.299 | 0.047 * | 0.122 # | 0.594 |
| Total V of the swing [mm/s] | 10.5 | 2.6 | 10.3 | 3.0 | 10.7 | 2.4 | 11.3 | 2.5 | 0.915 | 0.346 | 0.029 | |
| Sway V A-P [mm/s] | 8.8 | 2.4 | 8.9 | 3.0 | 9.1 | 2.4 | 9.8 | 2.5 | 0.663 | 0.422 | 0.021 | |
| Sway V M-L [mm/s] | 4.1 | 0.9 | 3.6 | 0.7 | 4.0 | 0.5 | 4.0 | 0.6 | 4.269 | 0.047 * | 0.121 # | 0.590 |
| The total area of swing, [mm$^2$] | 192.5 | 79.2 | 166.3 | 77.3 | 194.3 | 65.6 | 202.8 | 69.2 | 3.100 | 0.088 | 0.091 # | 0.588 |
| Sway area A-P [mm $\times$ s] | 79.0 | 36.8 | 72.4 | 27.0 | 74.8 | 17.8 | 71.2 | 19.1 | 0.000 | 0.984 | 0.000 | |
| Sway area M-L [mm $\times$ s] | 19.4 | 8.0 | 19.4 | 10.6 | 17.5 | 6.5 | 20.1 | 0.77 | 0.766 | 0.388 | 0.024 | |
| | | | | (Close Eyes) | | | | | | | | |
| The total length of the swing [mm] | 271.5 | 82.4 | 230.9 | 74.6 | 257.3 | 80.4 | 288.6 | 95.9 | 3.485 | 0.071 | 0.101 # | 0.537 |
| Sway path A-P [mm] | 237.6 | 76.3 | 200.7 | 73.2 | 226.2 | 78.8 | 259 | 92.6 | 3.708 | 0.063 | 0.107 # | 0.585 |
| Sway path M-L [mm] | 91.1 | 24.3 | 79.2 | 16.9 | 85.6 | 17.2 | 87.5 | 19.3 | 1.458 | 0.243 | 0.057 | |
| Total V of the swing [mm/s] | 13.6 | 4.1 | 11.5 | 3.7 | 12.9 | 4.0 | 14.4 | 4.8 | 3.548 | 0.069 | 0.103 # | 0.585 |
| Sway V A-P [mm/s] | 11.9 | 3.8 | 10.0 | 3.7 | 11.3 | 3.9 | 12.9 | 4.6 | 3.688 | 0.064 | 0.106 # | 0.584 |
| Sway V M-L [mm/s] | 4.6 | 1.2 | 4.0 | 0.8 | 4.3 | 0.9 | 4.4 | 1.0 | 2.666 | 0.113 | 0.079 # | 0.573 |
| The total area of swing [mm$^2$] | 271.2 | 141.1 | 186.7 | 83.2 | 216.8 | 93.3 | 269.1 | 142.6 | 5.605 | 0.024 * | 0.153 ## | 0.601 |
| Sway area-M-L [mm $\times$ s] | 85.7 | 30.5 | 69.7 | 25.2 | 73.4 | 21.3 | 74.5 | 15.6 | 0.247 | 0.623 | 0.008 | |
| Sway area A-P [mm $\times$ s] | 20.7 | 8.5 | 21.3 | 11.8 | 17.1 | 5.2 | 19.9 | 7.6 | 0.224 | 0.639 | 0.007 | |

Note: ICMT—interactive cognitive-motor training; CG—control group; M $\pm$ SD—Mean $\pm$ Standard Deviation; V—velocity; A-P—anterior-posterior; M-L—medial-lateral; *—$p < 0.05$; #—medium effect sizes; ##—large effect sizes.

For open-eye static balance, compared to the CG, there was a significant improvement in the total area of swing ($F = 5.605$, $p = 0.024$, $\eta_p^2 = 0.153$) for the ICMT groups of participants, with a large effect size. The total length of the swing ($F = 3.485$, $p = 0.071$, $\eta_p^2 = 0.101$), the sway path A-P ($F = 3.708$, $p = 0.063$, $\eta_p^2 = 0.107$), the total V of the swing($F = 2.181$, $p = 0.124$, $\eta_p^2 = 0.083$), the sway V A-P($F = 3.688$, $p = 0.064$, $\eta_p^2 = 0.106$), and the sway V A-P($F = 2.666$, $p = 0.113$, $\eta_p^2 = 0.079$) in the ICMT group were improved, but there was no significant difference; there was a medium effect size.

Table 3 shows the differences between the two groups in the effects of different exercise styles on LOS and postural stability in older adults. After 12 weeks of intervention compared to the control group, there was a significant improvement in the L angle and

a large effect size (F = 6.202, $p$ = 0.018, $\eta_p^2$ = 0.167) for the ICMT group. There was an improvement in the L angle, with a medium effect sizes (F = 2.404, $p$ = 0.131, $\eta_p^2$ = 0.072). For postural stability, we found an improvement in A-P stability (F = 2.952, $p$ = 0.096, $\eta_p^2$ = 0.087) in the ICMT group compared to the control group, but there was no significant difference with medium effect sizes.

**Table 3.** Differences between two groups in the effects of different exercise styles on LOS and postural stability in older adults.

| | ICMT ($n$ = 22), M $\pm$ SD | | | | CG ($n$ = 16), M $\pm$ SD | | | | Comparison of Two Groups | | | |
| --- | --- | --- | --- | --- | --- | --- | --- | --- | --- | --- | --- | --- |
| | Pre | | Post | | Pre | | Post | | F | $p$ | $\eta_p^2$ | 1-$\beta$ |
| *Limits of stability* | | | | | | | | | | | | |
| Average angle [°] | 6.2 | 0.7 | 6.1 | 0.6 | 6.0 | 0.6 | 6.0 | 0.6 | 0.948 | 0.338 | 0.030 | |
| Anterior angle [°] | 7.1 | 1.1 | 7.1 | 0.8 | 7.0 | 0.8 | 7.1 | 1.0 | 0.000 | 0.995 | 0.000 | |
| P angle [°] | 4.0 | 0.7 | 3.7 | 1.0 | 3.8 | 0.6 | 3.4 | 0.9 | 1.459 | 0.236 | 0.045 | |
| M angle [°] | 6.5 | 0.6 | 6.6 | 0.7 | 6.5 | 1.2 | 6.6 | 0.8 | 0.011 | 0.918 | 0.000 | |
| L angle [°] | 6.5 | 1.0 | 6.5 | 0.6 | 6.4 | 0.6 | 5.9 | 1.2 | 6.202 | 0.018 * | 0.167 ## | 0.606 |
| M-A Angle [°] | 7.7 | 1.2 | 7.8 | 0.9 | 7.7 | 1.0 | 8.0. | 1.0 | 0.349 | 0.559 | 0.011 | |
| L-A Angle [°] | 7.7 | 1.4 | 7.8 | 0.6 | 7.4 | 0.8 | 7.5 | 1.2 | 2.404 | 0.131 | 0.072 # | 0.570 |
| M-P Angle [°] | 5.0 | 0.7 | 4.8 | 0.9 | 4.8 | 0.9 | 4.9 | 1.0 | 0.000 | 0.987 | 0.000 | |
| L-P Angle [°] | 5.0 | 0.7 | 4.7 | 1.1 | 4.7 | 0.8 | 4.7 | 0.8 | 0.349 | 0.559 | 0.011 | |
| *Postural stability* | | | | | | | | | | | | |
| Stability Overall | 0.8 | 0.2 | 0.8 | 0.2 | 0.8 | 0.3 | 0.8 | 0.2 | 0.379 | 0.542 | 0.012 | |
| A-P | 0.6 | 0.3 | 0.5 | 0.2 | 0.4 | 0.3 | 0.4 | 0.2 | 2.952 | 0.096 | 0.087 # | 0.578 |
| M-L | 0.6 | 0.1 | 0.6 | 0.2 | 0.6 | 0.2 | 0.6 | 0.2 | 0.068 | 0.796 | 0.002 | |
| Sway Overall | 0.3 | 0.4 | 0.2 | 0.3 | 0.5 | 1.1 | 0.3 | 0.5 | 0.632 | 0.433 | 0.020 | |
| A-P | 0.2 | 0.3 | 0.2 | 0.4 | 0.4 | 0.9 | 0.3 | 0.4 | 0.082 | 0.777 | 0.003 | |
| M-L | 0.2 | 0.3 | 0.2 | 0.2 | 0.3 | 0.6 | 0.2 | 0.2 | 0.319 | 0.576 | 0.010 | |

Note: ICMT—interactive cognitive-motor training; CG—control group; M $\pm$ SD—Mean $\pm$ Standard Deviation; V—velocity; A-P—anterior-posterior; M-L—medial-lateral; *—$p$ < 0.05; #—medium effect sizes; ##—large effect sizes.

We found a significant improvement in the sway path in the M-L direction in the ICMT group in the eyes-open state and the total sway path in the eyes-closed state ($p$ < 0.05, Figure 8).

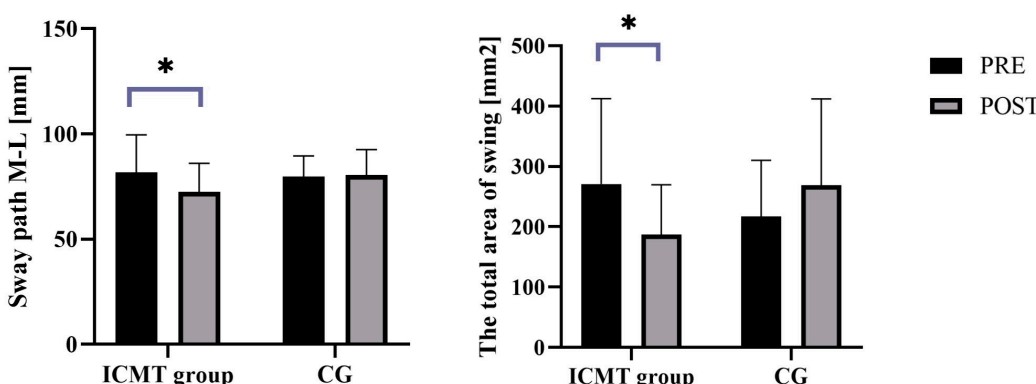

**Figure 8.** Before-and-after plot of the difference between the open-eye oscillation path and the total area of closed-eye oscillation in the two groups. *—$p$ < 0.05.

## 5. Discussion

This study is a randomized controlled comparison that examines the effects of ICMT on the balance in older adults who live in the community. We hypothesized that an ICMT intervention based on Nintendo Switch gaming would improve balance in older adults. The results showed that there was a significant difference between the two groups regarding the effect of interventions and 12 weeks of ICMT on improving balance in older adults. ICMT had a more pronounced improvement in static balance in older adults with open and

closed eyes, and this improvement was not detected in the CG. ICMT also had a significant improvement in degrees of LOS in the L angle in dynamic balance. Our study substantiates the effectiveness of ICMT as a form of physical exercise in enhancing balance in older adults while also offering novel ideas and approaches for physical activity among older adults. It highlights the potential of ICMT as an innovative and effective approach for improving balance in older adults.

ICMT improved static balance in older adults. This is similar to the results of previous studies [34]. A study has found that the amplitude of oscillation in the M-L direction is a predictor of fall risk in older adults [35], and our study found significant enhancements in the path and speed of sway in the M-L direction through ICMT, further supporting previous research. Another study exploring the Nintendo Wii Fit intervention found positive balance impacts in women aged 30–58 years [36]. That may be due to the ability of ICMT to enhance proprioception in older adults [37,38]. We use games that involve both the upper and lower limbs. There is evidence that muscle strength is associated with balance [39,40]. A physiologic response to exercise can improve neuromuscular and musculoskeletal health [41,42]. A randomized controlled trial found that seated boxing improved upper limb strength, balance, and gait in stroke patients [43]. Aerobic boxing requires the participant to control the COG to prevent the body from swinging out of balance with the upper limbs [44,45]. At the same time, our game required participants to make immediate judgments about the icons on the screen, and the receptors in the handle also scored the participants' movements, requiring more sensory involvement in the decision-making process. The additional cognitive intervention in ICMT may have contributed to the better outcome of the intervention in ICMT [46]. There are two studies where they found that Wii Fit exercise games were beneficial for balance in older adults and also boosted their balance confidence [47,48]. We know that balance is linked to cognitive function. ICMT is an intervention in which movement and cognition work together and can improve balance.

In terms of dynamic balance, ICMT has shown some improvement in the LOS in older adults. The most important finding of our study was that ICMT significantly improved LOS in the L angle in older adults compared to the control group. A study has shown that dance focuses more on dynamic balance skills [46]. The ZUMBA dance is available in our selection of games. Our study can also be considered dual-task training because older adults had some cognitive tasks when exercising, such as judging dance movements and punching styles. Training plantar flexor muscle strength may improve the LOS [49]. Some studies suggest that the control of COG is related to sensory function [50]. A similar randomized controlled trial using a virtual reality system (the Balance Rehabilitation Unit [BRU]) to train 30 older adults has shown that training improved LOS and COP in older adults [50]. Previous studies have confirmed that ICMT can reduce postural sway [51–54]. A study has found Tai Chi can improve postural stability in patients with Parkinson's disease [55]. Postural control in older adults is a multifaceted process that involves the interaction of sensory and motor systems, enabling the perception of environmental cues, adaptation to changes in body orientation, and the maintenance of the body's center of gravity within the base of support. In ICMT, participants must continuously assess their movements and make real-time adjustments based on auditory and visual feedback. This intervention progressively intensifies by increasing task difficulty, enhancing both motor performance and balance [56]. Consequently, these enhancements in motor skills and balance may extend to an overall improvement in participants' postural control.

Our study provides the first empirical evidence that ICMT based on Nintendo Switch games can improve balance abilities in older adults. Additionally, we offer a comprehensive and objective evaluation of ICMT's effectiveness in enhancing static and dynamic balance. Unlike conventional ICMT methods, the integration of Nintendo Switch games introduces a distinctive dimension of realism through their advanced visual effects and immersive experiences. This immersion potentially augments the engagement and commitment of participants throughout the training process. Furthermore, the portability and multiplayer

functionality intrinsic to Nintendo Switch games foster social interaction and engagement among older adults, fostering a sense of camaraderie that can significantly contribute to the overall training experience. Our findings underline the value of ICMT as an effective intervention for enhancing balance and bridging the cognitive and motor domains in a concurrent task framework. This unique integration enables participants to engage in dual-task scenarios, enhancing their cognitive abilities and motor skills. The tailored progression inherent in ICMT augments adherence and motivation, thus promoting sustained engagement in the program. ICMT is an innovative approach with versatile applications spanning healthcare, wellness initiatives, community centers, and rehabilitation facilities. The potential implications of ICMT extend beyond balance enhancement, as it presents a holistic strategy for nurturing physical and cognitive health. While our investigation spotlights the efficacy of ICMT through Nintendo Switch games, future research should delve into its broader effects on additional physical functions. With its dual-task paradigm, personalized progression, and social dynamics, ICMT bears the potential to reshape the landscape of healthcare interventions, promoting holistic well-being in aging populations.

ICMT is an effective tool for rehabilitation and health promotion, providing notable benefits to older adults through enhancing balance abilities. By effectively reducing the risk of falls and associated injuries, ICMT contributes to optimizing healthcare resources, aligning with the goals of disease prevention and sustainable development [57,58]. Additionally, beyond the physical advantages, ICMT also facilitates social interaction. Participation in ICMT not only exercises the body but also fosters social engagement and promotes overall well-being among older adults [17]. This social aspect plays a crucial role in their sustainable development, as it not only enhances the overall quality of life but also extends the years of healthy and active living.

## 6. Strengths and Limitations

Our study has several strengths: (1) We incorporated both static and dynamic balance into our research design while controlling for a plethora of factors and employing more objective measurements to comprehensively evaluate the impact of ICMT on balance in older adults. (2) This study was a randomized intervention study with a control group. (3) The strict control of exercise intensity through objective heart rate monitoring during the intervention ensures the safety of the intervention in older adults.

Some limitations should be considered when interpreting the findings of this study: (1) Firstly, the sample size was small, the study was conducted over a relatively short period, and no active control group was set up, which could limit the generalizability of the findings. (2) Secondly, there was a significant difference in the sex distribution between male and female participants, which could have potentially influenced the findings. (3) Thirdly, the study did not rigorously control participant diet or medication during the intervention. In the future, a larger sample size and a long-term follow-up study of participants should be conducted to investigate the enduring benefits of ICMT. A comparative analysis of the effects of ICMT and conventional physical training may be conducted to investigate the disparities between the two. Cognitive function could be studied as a predictor of a lessened decline in balance in older adults.

## 7. Conclusions

ICMT significantly improves balance in older adults, demonstrating remarkable efficacy in both static and dynamic balance. It is an effective intervention that enhances balance ability in older adults and potentially provides lasting health benefits. This study contributes to the application and advancement of technology in sustainable development by offering solutions to enhance the health and well-being of older adults. The objective of this research is to provide evidence of ICMT's efficacy and its potential as a valuable intervention for older adults, ultimately improving their balance and overall quality of life.

**Author Contributions:** Conceptualization, W.Z.; methodology, J.Z.; formal analysis, J.G.; investigation, L.Z. (Ling Zhang).; data curation, F.G.; writing—original draft preparation, L.Z. (Longhai Zhang).; writing—review and editing, S.Y. and Y.L.; funding acquisition, W.Z. All authors have read and agreed to the published version of the manuscript.

**Funding:** This study was supported by the MOE (Ministry of Education in China) Project of Humanities and Social Sciences (20YJC890053) and the Shaanxi Province Social Science Foundation Program (2020Q009).

**Institutional Review Board Statement:** The study was conducted according to the guidelines of the Declaration of Helsinki and approved by the Academic Committee of Shaanxi Normal University (202116003; 15 July 2021).

**Informed Consent Statement:** Informed consent was obtained from all participants involved in the study.

**Data Availability Statement:** The data presented in this study are available upon request from the corresponding author. The data are not publicly available due to confidentiality.

**Acknowledgments:** We thank all the funds for their support and the valuable contributions of all investigators and participants.

**Conflicts of Interest:** The authors declare no conflict of interest.

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
