# Peer review of "Interactive Cognitive Motor Training: A Promising Approach for Sustainable Improvement of Balance in Older Adults"

_sustainability, doi:10.3390/su151813407_

Round 1
Reviewer 1 Report (Previous Reviewer 3)
The authors have made sufficient revisions to this manuscript. We can accept it in the present version.
Author Response
Thank you very much for your review.
Reviewer 2 Report (New Reviewer)
Study results should be given clearly.
Strengths and weaknesses of the study (6.) should be given in the Method section.
How much time each of the three exercises was done in a 45-minute period should be given. Very few people smoke and drink alcohol. These parameters can be removed. The subject is very messy.
Author Response
Please see the attachment.

Reviewer 3 Report (New Reviewer)
Regarding the evaluation of the manuscript entitled "Interactive Cognitive Motor Training: A Promising Approach 2 for Sustainable Improvement of Balance in Older Adults"
· Please describe how was the sample size calculated?
· Could you explain more about exercise sessions design for ICMT group through (intensity, volume, repetition, sets, and rest) especially between male and female?
· Page 2, Lines 75: The Physical Activity Readiness Questionnaire (PAR-Q). Could you elaborate more in discussion?
· Page 12, Lines 369-370: Please clarify the point you are attempting to make. "Future research should explore its effects on other physical functions and long-term sustainability".
· The idea behind this work is good but discussion needs precision and clarity. Could you elaborate more on this by specifically talking about ICMT based on Nintendo Switch games can improve balance abilities in older adults.
Minor editing of English language required
Author Response
Please see the attachment.

This manuscript is a resubmission of an earlier submission. The following is a list of the peer review reports and author responses from that submission.
Round 1
Reviewer 1 Report
Abstract
Headings should be removed.
Abstract should be a single paragraph of about 200 words maximum. I think you can shorten the background.
You should also state what’s the gap that you’re trying to fill.
keywords usually should be different from that used in the main title.
Introduction
it is not clear whether you are considering AVG and ICMT the same thing.
The objective of your paper, in my opinion, is not entirely clear, especially between what is stated in the abstract and what is stated in the introduction section.
Abstract:
"The primary objective of this research is to enhance the physical health of older adults, thus mitigating fall-related injuries and fostering sustainable social well-being."
Introduction:
"In this study, we conducted a 12-week randomized controlled trial to investigate the long-term effect of ICMT on balance in older adults. The research primarily aims to examine the potential of AVG (Active Video Games) in enhancing balance capabilities among older adults, thereby mitigating the risk of falls and related injuries".
AVG: you should state it’s meaning the first time you use the abbreviation, on line 63, by putting the abbreviation in brackets.
Methods
The methods section is sufficiently well described.
Validity of the findings
The results and discussion section are quite clear and organised; however several studies have already observed that AVG and ICMT are able to improve balance in the elderly so, in my opinion, it is not yet clear what is the gap you are trying to fill with your work, you should state it.
Minor editing of English language required
Reviewer 2 Report
GENERAL COMMENTS
The research design that compares an experimental group that performs any intervention with a control group that "continued their daily activities without modification" is too basic.
The procedure is very confusing and the study design is unclear and it was not conducted with academic scientific rigor and this does not allow the replicability of the study.
ABSTRACT
I suggest making the background shorter (it will be argued in the introduction section of the main manuscript).
Add some details about the type of intervention. For example, type of exercises? (aerobic, balance); was there a specific intensity expected?
Among the results rather than indicating a "significant difference" it would be more appropriate to report a "significant improvement" because this indicates what kind of difference it was.
INTRODUCTION
The meaning of the acronym AVG should be clearly reported the first time it appears in the text (page 2 line 63)
Authors failed to report the hypothesis of the study.
MATERIALS AND METHODS
Recruitment process details are needed.
Authors should report the randomization method used.
Only inclusion criteria were reported. Were there no exclusion criteria?
The authors first stated that they recruited 60 participants. They later reported that only 56 met the inclusion criteria. This passage is not clear. Recruiting is different from enrollment. Moreover, in the flow diagram authors explain why they excluded 4 subjects reporting also that 1 was excluded for “hearth disease” but this criteria had not been mentioned in the manuscript.
(Page 2 lines 86,87): “… 10 participants withdrew from the intervention for medical reasons (ICMT group=4, CG =6) and 6 participants withdrew from the intervention for time conflicts (ICMT group=2, CG =6) …” What do authors mean by medical reasons and time conflicts? Too generic to categorize dropouts in this way.
Was the sample size power calculated a priori?
Which version of the Physical Activity Readiness Questionnaire (PAR-Q) was used? I think a version in the native language of the participants. If so, it should be specified and the reliability of the instrument should be reported.
The paragraph “Intervention” is too general.
Authors stated that "Participants were asked to wear the accelerometer on their non-dominant wrist for seven consecutive days". However the intervention lasted 12 weeks. When was the accelerometer actually worn? And according to what criteria was one week established rather than another? This is a fundamental step in the research design of this study.
It is unclear when the measurements were taken. In addition, details of the data collection settind should be reported to allow replicability of the study.
Please explain why the balance test lasted 20 seconds. 51.2 seconds is common used in the literature. Moreover, explain why the recovery interval lasted from 30 to 60 seconds.
Authors stated that each session lasted 60 minutes, incorporating a 10-minute warm-up and 5-minute cool-down period. Hence, 45 minutes of central phase? However, in the paragraph “Interactive cognitive-motor training” they referred to “one-hour group sessions”.
Participants were ensured to exercise at a moderate to vigorous intensity (exercise intensity = 65 - 75% HRmax). Please add more details. Was this only recorded during the central phase? How did they maintain this intensity? Has a certain level of games been established a priori?
Authors should report validity and reliability of the Heart Rate System used. Similarly for the game device and for the instruments used for the evaluation of both static and dynamic balance.
Minor editing of English language required.
Reviewer 3 Report
I have reviewed the manuscript titled “Interactive Cognitive Motor Training: A Promising Approach for Sustainable Improvement of Balance in Older Adults”. The manuscript reports a randomized controlled trial that investigated the effects of interactive cognitive-motor training (ICMT) based on Nintendo Switch games on the balance of older adults living in the community. The authors found that ICMT improved static balance and maintained dynamic balance in older adults compared to a control group. The authors suggest that ICMT is an effective and enjoyable intervention for enhancing balance and reducing fall risk in older adults, and that it has potential benefits for sustainable health and social well-being.
Overall, I think the manuscript is well-written, clear, and informative. The research topic is relevant and timely, as population aging poses significant challenges for health care systems and society. The research question is well-defined and motivated by previous literature. The methods are appropriate and rigorous, and the results are presented in a logical and comprehensive manner. The discussion is balanced and insightful, and the conclusions are supported by the data.
However, I have some comments that I think the authors should address before the manuscript can be accepted for publication. Below are my specific comments and suggestions for improvement.
Introduction
1. The sentences from lines 70 to 74 should be reorganized to improve their logic and coherence. For example, the authors could start with the definition and prevalence of balance problems in older adults, then explain the consequences and risk factors of falls, and finally introduce the potential benefits of ICMT for balance improvement.
2. The research gaps should be explicitly highlighted by mentioning the lack of previous research or gaps in the introduction, which would set the tone for the study. For example, the authors could state what is known and unknown about ICMT for balance in older adults, and how their study addresses the existing knowledge gaps.
Participants and methods
3. The author mentions smoking and alcohol consumption in the article, but it is unclear how these factors affect the balance. The author should provide some evidence or rationale for including these factors in the analysis, or exclude them if they are not relevant.
4. What are the specific advantages of interactive cognitive-motor training over sports games in this study? The author should explain why they chose Nintendo Switch games as the intervention, and how they differ from other types of physical exercise or games in terms of cognitive and motor demands, engagement, and feedback.
5. With many tests available on Biodex Balance Systems, why were these two tests specifically chosen? The author should justify their choice of tests for measuring dynamic balance, and how they relate to the research question and objectives.
Conclusions
6. The conclusion section should streamline the statement instead of restating the results. The author should focus on summarizing the main findings, and implications of the study, and suggest directions for future research.
The quality of English language in the manuscript is generally good, but there are some minor errors and inconsistencies that need to be corrected. Here are some examples:
On page 1 line 12, “with aging presenting numerous negative effects on society” should be “with aging presenting numerous challenges for society”.
On page 1 line 12-13, “The various risks associated with the aging process pose a significant threat to older adults’ physical and mental health” should be “Various risks associated with aging pose significant threats to older adults’ physical and mental health”.
On page 1 line 24-25, “The intervention group engaged in 60-minute exercise sessions three times a week for 12 weeks” should be “The intervention group engaged in 60-minute exercise sessions three times per week for 12 weeks”.
On page 3 line 112, “The following are the three games selected for this study” should be “The following three games were selected for this study”.
On page 6 line 165, “The participant completed the test three times with eyes open and closed for 20 seconds each” should be “The participant performed the test three times with eyes open and closed for 20 seconds each”.
On page 5 line 148, “The metrics tested in this study include” should be “The metrics tested in this study included”.
On page 7 line 196-197, “The test metrics are average angle” should be “The test metrics were average angle”.
I suggest that the authors proofread their manuscript carefully to ensure that the language is clear and accurate.
Reviewer 4 Report
You will find here some comments to consider for a better version:
1. Provide a theoretical base and empirical research on the subject.
2.- Insert an explicit description of the results is required, if possible with graphical representations (such as scatter plots or a whisker diagram), so that future readers can have a better understanding.
3.- Provide a separate discussion section with a broader view of how this approach can be used beyond the specific study.
4.- Restructure the conclusion indicating at least the main contribution, what objective(s) has/have been achieved, argue
5. - Several references should be updated (5 yeast later), in particular those related to the main approach.
Round 2
Reviewer 1 Report
the authors followed up on my comments. I have no further comments.
Reviewer 2 Report
Although the authors addressed some of my earlier doubts, my main one persists: the study design is too basic. Of course a comparison between two groups of which one performs a physical intervention and one that does not carry out any physical protocol can, eventually, lead to positive changes only to the active group. But this is widely known.
"set up a blank control group because the study was conducted during the period of noncoronary pneumonia" is not a reasonable reason for academic soundness. Where is the novelty of the study? Previous studies have already investigated the positive effects of various cognitive-motor interventions.
Moreover, what authors mean for "continued their daily activities without modification"? This aspect does not inform the amount of physical activity they practiced. Was this group homogeneous? Or were there both active and inactive subjects in it?
I appreciate the efforts of the authors, and suggest this could be a starting point. However, but as confirmed by themselves, it would be appropriate to conduct further studies to improve their experiment.
None
Reviewer 4 Report
This time my questions were already answered satisfactorily